# Research on Energy-Saving Routing Technology Based on Deep Reinforcement Learning

**Xiangyu Zheng** [1] , **Wanwei Huang** [1,*], **Sunan Wang** [2], **Jianwei Zhang** [1] and **Huanlong Zhang** [1]

1   College of Software Engineering, Zhengzhou University of Light Industry, Zhengzhou 450001, China; 332013020641@email.zzuli.edu.cn (X.Z.); zhangjw@zzu.edu.cn (J.Z.); hlzhang@zzuli.edu.cn (H.Z.)
2   School of Electronic and Communication Engineering, Shenzhen Polytechnic, Shenzhen 518055, China; wangsunan@szpt.edu.cn
*   Correspondence: 2016044@zzuli.edu.cn

**Abstract:** With the vigorous development of the Internet, the network traffic of data centers has exploded, and at the same time, the network energy consumption of data centers has also increased rapidly. Existing routing algorithms only realize routing optimization through Quality of Service (QoS) and Quality of Experience (QoE), which ignores the energy consumption of data center networks. Aiming at this problem, this paper proposes an Ee-Routing algorithm, which is an energy-saving routing algorithm based on deep reinforcement learning. First, our method takes the energy consumption and network performance of the data plane in the software-defined network as the joint optimization goal and establishes an energy-efficient traffic scheduling scheme for the elephant flows and the mice flows. Then, we use Deep Deterministic Policy Gradient (DDPG), which is a deep learning framework, to achieve continuous and energy-efficient traffic scheduling for joint optimization goals. The training process of our method is based on a Convolutional Neural Network (CNN), which can effectively improve the convergence efficiency of the algorithm. After the algorithm training converges, the energy-efficient path weights of the elephant flows and the mice flows are output, and the balanced scheduling of routing energy-saving and network performance is completed. Finally, the results show that our algorithm has good convergence and stability. Compared with the DQN-EER routing algorithm, Ee-Routing improves the energy saving percentage by 13.93%, and compared with the EARS routing algorithm, Ee-Routing reduces the delay by 13.73%, increases the throughput by 10.91%, and reduces the packet loss rate by 13.51%.

**Keywords:** software-defined network (SDN); energy-efficient routing; deep deterministic policy gradient (DDPG); convolutional neural network (CNN)

## 1. Introduction

With the rapid development of the Internet, global data center traffic has exploded. The data center networks carry thousands of services, and the service traffic demand is unevenly distributed and dynamic changes are large. As a result, the data center networks are facing a huge energy consumption problem [1]. Existing research shows that in recent years, data center networks' energy consumption accounts for 8% of global electricity consumption, of which network infrastructure energy consumption accounts for 20% of data center energy consumption [2]. Facing the ever-complex and ever-changing network application services and the sharp increase in the energy consumption of network infrastructure, the traditional routing algorithms only focus on network Quality of Service (QoS) [3] and user Quality of Experience (QoE) [4], and they have been unable to better meet application requirements. Network energy-saving technology is also a goal that needs to be guaranteed and optimized [5]. Therefore, under the premise of ensuring network service requirements, it is of great significance to study energy-saving optimization technologies for data center networks to reduce network energy consumption.

With the rise of artificial intelligence, researchers have carried out in-depth research on network performance optimization and proposed a series of intelligent routing algorithms for network performance optimization [6]. In reference [7], an intelligent flow control method based on deep learning is proposed. This method selects a near-optimal routing strategy according to the degree of link congestion by inputting a specific state into a convolutional neural network. Compared with the traditional routing algorithm, it achieves a low packet loss rate and average delay, but this method usually needs to obtain a large number of correctly labeled data sets in the network, and the data labeling process still requires manual participation. Therefore, the intelligent routing algorithm based on supervised learning is applied, subject to certain restrictions. In reference [8], an intelligent routing algorithm based on deep reinforcement learning is proposed, and with the help of SDN, it dynamically collects network traffic distribution information, schedules routing strategies in time, and realizes end-to-end delay optimization under different throughputs. In reference [9], an intelligent routing method based on deep reinforcement learning in data center networks is proposed. This method realizes adaptive routing optimization under different network states through multiple network resource reorganization methods. Compared with OSPF and TIDE, it reduces the delay and improves the load balancing. In reference [10], a constrained intelligent routing method based on deep reinforcement learning is proposed, which solves the constraint problem through the Lagrangian multiplier method so that the routing service can meet the differentiated needs of users for network performance. In reference [11], a multi-path routing algorithm based on link real-time status and traffic characteristics is proposed. This method routes the elephant flows and the mice flows through the proportion of link weights and optimizes the network performance indicators such as average link utilization and throughput. In reference [12], an intelligent-driven network architecture based on SDN is proposed. This method takes the network delay and throughput in the data plane as the optimization goal and effectively improves the network load balancing compared with the traditional routing algorithms OSPF and ECMP.

However, the above methods do not consider the problem of network energy consumption. In reference [13], a DQN-based network resource allocation algorithm is proposed, which adopts the resource allocation strategy of the Deep Q Network, and adopts experience repetition and the target network. To overcome the instability and divergence of results caused by previous network states, the algorithm can maximize the overall throughput of the network while making the network more energy-efficient and stable, however, DQN is usually used to deal with discrete action problems, which are difficult to deal with continuously. Therefore, the energy-saving effect of this method still needs to be improved. In reference [14], an SDN-based intelligent network energy consumption optimization method is proposed. The method adjusts the activation and sleep of network devices through cooperative sleep technology to reduce network energy consumption. Compared with the traditional routing algorithm, the energy-saving effect is effectively improved. However, it increases the load pressure on the control plane during the process of adjusting device activation and sleep. On this basis, in reference [15], an energy-saving topology optimization algorithm for control plane performance optimization is proposed. This method adapts the control plane load and data plane energy consumption through the design of traffic awareness and device sleep technology so that obtaining an energy-saving topology improves the performance of the control plane to a certain extent, but it will cause a certain delay consumption when the switch frequently changes between device activation and sleep. To sum up, intelligent routing algorithms based on machine learning have shown good performance advantages in data center networks. However, the above algorithms usually only consider network performance optimization or relatively simple network energy-saving optimization and do not establish joint optimization goals for energy-saving and network performance. Under the circumstance that the network scale is constantly expanding and the network traffic demand is constantly complex, it is difficult for such intelligent routing algorithms to formulate specific energy-efficient traffic optimization

goals for elephant flows and mice flows, which often leads to low efficiency of routing algorithms, and performance indicators such as network energy consumption, average end-to-end delay, throughput, and packet loss rate still need to be improved. In addition, the above-mentioned intelligent routing algorithms usually use traditional algorithm frameworks and neural networks for training, and the processing speed of complex data with multiple dimensions is slow, and the convergence and effectiveness of the algorithms need to be improved.

In response to the above problems, based on the Software-Defined Network (SDN) [16] technology, this paper takes the energy-saving and network performance of the data plane as the joint optimization goal and establishes the energy-saving and network performance optimization models of elephant-flow and mice-flow scheduling, and proposes an Ee-Routing algorithm, which is an energy-saving routing algorithm based on deep reinforcement learning, uses an improved Deep Deterministic Policy Gradient (DDPG) [17] as a deep reinforcement learning framework, which can achieve continuous traffic scheduling and optimization of the joint goal. The training process is based on a Convolutional Neural Network (CNN) [18], taking advantage of its local perception and parameter sharing, and ensuring that the convolution kernel has a strong response to the spatial local pattern of the input, and can input high-dimensional vectors into the network at the same time, avoiding the complexity of data reconstruction in the process of feature extraction and classification. In this algorithm, the state features of the data plane are input into the CNN, and based on the joint optimization goal of energy-saving and network performance of the data plane, high-efficiency traffic scheduling is realized. The contributions of this paper are summarized as follows:

(1) We analyze that the existing routing algorithms are difficult to deal with the balance between energy-saving and network performance in the data plane, and propose an intelligent routing algorithm Ee-Routing that jointly optimizes the energy-saving and network performance.
(2) The traffic scheduling optimization goals of elephant flows and mice flows are established, a DDPG algorithm framework suitable for improving energy-saving and network performance advantages is proposed, and the CNN structure is adapted for the algorithm convergence efficiency.
(3) Using Fat Tree as the network topology, the convergence, energy-saving, and network performance advantages of the Ee-Routing algorithm are verified under different traffic intensities.

This paper mainly consists of six sections. In Section 2, the energy-saving routing optimization goals are established, and in Section 3 the energy-saving network architecture based on SDN is built. In Section 4 we implemented the Ee-Routing, an energy-saving routing algorithm based on DDPG, and in Section 5 we verified the energy-saving and network performance advantages of Ee-Routing through experimental comparison. Finally, Section 6 presents our conclusion.

## 2. Modeling of Energy-Efficient Routing

Traditional network traffic scheduling methods in data centers usually use a unified traffic scheduling method, which will inevitably lead to problems such as low real-time scheduling, unbalanced resource allocation, and high energy consumption [19]. In order to ensure the balance of traffic in user services, our method further divides traffic into elephant flows and mice flows for dynamic scheduling. The elephant flow usually has a long survival time and carries a large amount of data, occupying 80–90% of the total network traffic of the data center, the data traffic in less than 1% of the traffic packets can reach more than 90%, and less than 0.1% of the data traffic can last for 200 s [20]. The mice flow usually has a short survival time and carries a small amount of data, the total number reaches 80% of the total flows, and the transmission time is within 10 s [21]. Therefore, according to the characteristics of different types of traffic, our method establishes different optimization

methods for the elephant flows and the mice flows so as to realize the energy-saving scheduling of the elephant flows and the mice flows.

　　In this paper, we assume that energy-efficient traffic scheduling is performed when the data center network topology has been determined and both links and switch activation and sleep are well-defined. On this basis, the network energy consumption model can be simplified to the energy consumption model of link rate level [22], and the link power consumption function is denoted as $Power(r_e)$, where $r_e(t)$ is the link transmission rate, and the calculation process is shown in Equation (1).

$$Power(r_e)=\sigma+\mu r_e^{\alpha}(t), \quad 0 \leq r_e \leq \beta R \tag{1}$$

　　In Equation (1), $\sigma$ represents the energy consumption when the link is idle, $\mu$ represents the link rate correlation coefficient, $\alpha$ represents the link rate correlation index and $\alpha > 1$, so that $(r_{e1} + r_{e2})^{\alpha} > r_{e1}^{\alpha} + r_{e2}^{\alpha}$, $r_{e1}$ and $r_{e2}$ are the link transmission rates of the same link at different times or different links, $Power(\cdot)$ can be added, $\beta$ is the link redundancy parameter and takes the value range (0, 1), and $R$ is the maximum transmission rate of the link. Therefore, from Equation (1), it can be seen that the minimized link energy consumption is achieved when the flow is transmitted uniformly in time and space. The total network energy consumption $Power_{total}$ during flow transmission is calculated as shown in Equation (2).

$$Power_{total}=\int_{p'_i}^{q'_i} \sum_{e \in E_a} (\sigma+\mu r_e^{\alpha}(t))dt, \quad r_e(t) = \sum_{e \in E_b} s_i(t) \tag{2}$$

　　In Equation (2), $p'_i$ and $q'_i$ respectively represent the start time and latest end time of the flow in the actual transmission process, $E_a$ represents the set of active links, and $E_b$ represents the set of all links that have flows on the link, and $s_i(t)$ is the transmission rate of a single flow.

　　We define the topology of the data center networks as $G = (V, E, C)$, where $V$ represents the nodes set of the network topology, $E$ represents the links set of the network topology, and $C$ represents the capacity set of each link. We assume that the set of elephant flows transmitted in the network topology are denoted as $Flow_{elephant} = \{f_m | m \in N^+\}$, the set of mice flows are denoted as $Flow_{mice} = \{f_n | n \in N^+\}$, where m is the number of elephant flows, n is the number of mice flows, $f_i = (s_i, d_i, p_i, q_i, r_i)$, and $s_i$ represent the source node of the flow, $d_i$ is the destination node of the flow, $p_i$ is the start time of the flow, and $q_i$ is the end time of the flow, $r_i$ represents the bandwidth requirement of the flow. In the network topology, the end-to-end delay is recorded as $delay(x)$, the packet loss rate is recorded as $loss(x)$, and the throughput is recorded as $throught(x)$. The average packet loss rate and average throughput of the elephant flows, the average end-to-end delay and the average packet loss rate of the mice flows are calculated as shown in Equations (3)–(6).

$$Loss_{elephent} = \frac{\sum\limits_{i=1}^{m} loss(f_m)}{m} \quad m \in N^+ \tag{3}$$

$$Throught_{elephent} = \frac{\sum\limits_{i=1}^{m} throught(f_m)}{m} \quad m \in N^+ \tag{4}$$

$$Delay_{mice} = \frac{\sum\limits_{i=1}^{n} delay(f_n)}{n} \quad n \in N^+ \tag{5}$$

$$Loss_{mice} = \frac{\sum\limits_{i=1}^{n} loss(f_n)}{n} \quad n \in N^+ \tag{6}$$

This method takes the energy-saving and the network performance of the data plane as the joint optimization goal for traffic scheduling, and the main optimization goals include: (1) the weighted minimum value of the network energy consumption and the average packet loss rate and the inverse of the throughput of the elephant flows; (2) the weighted minimum of network energy consumption and the average packet loss rate and average end-to-end delay of the mice flows. In order to simplify the calculation method, it is necessary to convert the quantified expressions into scalars so as to complete the normalization of data plane performance and energy-saving, and the calculation process is shown in Equations (7)–(11).

$$Power_{total}' = \frac{Power_{total_i} - \min\limits_{1 \leq j \leq m+n} \left\{ Power_{total_j} \right\}}{\max\limits_{1 \leq j \leq m+n} \left\{ Power_{total_j} \right\} - \min\limits_{1 \leq j \leq m+n} \left\{ Power_{total_j} \right\}} \tag{7}$$

$$Loss_{elephent}' = \frac{Loss_{elephent_i} - \min\limits_{1 \leq j \leq m} \left\{ Loss_{elephent_j} \right\}}{\max\limits_{1 \leq j \leq m} \left\{ Loss_{elephent_j} \right\} - \min\limits_{1 \leq j \leq m} \left\{ Loss_{elephent_j} \right\}} \tag{8}$$

$$Throught_{elephent}' = \frac{Throught_{elephent_i} - \min\limits_{1 \leq j \leq m} \left\{ Throught_{elephent_j} \right\}}{\max\limits_{1 \leq j \leq m} \left\{ Throught_{elephent_j} \right\} - \min\limits_{1 \leq j \leq m} \left\{ Throught_{elephent_j} \right\}} \tag{9}$$

$$Delay_{mice}' = \frac{Delay_{mice_i} - \min\limits_{1 \leq j \leq n} \left\{ Delay_{mice_j} \right\}}{\max\limits_{1 \leq j \leq n} \left\{ Delay_{mice_j} \right\} - \min\limits_{1 \leq j \leq n} \left\{ Delay_{mice_j} \right\}} \tag{10}$$

$$Loss_{mice}' = \frac{Loss_{mice_i} - \min\limits_{1 \leq j \leq n} \left\{ Loss_{mice_j} \right\}}{\max\limits_{1 \leq j \leq n} \left\{ Loss_{mice_j} \right\} - \min\limits_{1 \leq j \leq n} \left\{ Loss_{mice_j} \right\}} \tag{11}$$

$Power_{total}'$ represents the normalized value of the network energy consumption of the current flow, $Loss_{elephent}'$ represents the normalized value of the packet loss rate of the current elephant flow, $Throught_{elephent}'$ represents the normalized value of the throughput of the current elephant flow, $Delay_{mice}'$ represents The normalized value of the delay of the current mice flow, $Loss_{mice}'$ represents the normalized value of the packet loss rate of the current mice flow. After the normalization is completed, we establish the energy-saving and network performance optimization goals $\min \phi_{elephent}$ and $\min \phi_{mice}$ for elephant flows and mice flows scheduling, respectively, and the calculation process is shown in Equations (12) and (13).

$$\min \phi_{elephent} = \eta Power_{total}' + \tau Loss_{elephent}' + \rho \frac{1}{Throught_{elephent}'} \tag{12}$$

$$\min \phi_{mice} = \eta Power_{total}' + \tau Loss_{mice}' + \rho Delay_{mice}' \tag{13}$$

In Equations (12) and (13), $\eta$, $\tau$, and $\rho$ represent the weight parameters for energy-saving and network performance in the data plane, and $\eta$, $\tau$, and $\rho$ are all between 0 and 1. In order to ensure that the above traffic scheduling process is not affected by the environment, this method defines traffic transmission constraints as shown in Equations (14) and (15).

$$\int_{p'_i}^{q'_i} s_i(t)dt = c_i \tag{14}$$

$$\sum_{v \in \Gamma(u)} \left( f_i^{uv} - f_i^{vu} \right) = \left\{ \begin{array}{ll} c_i, & if \ u = s_i \\ -c_i, & if \ u = d_i \\ 0, & else \end{array} \right\} \tag{15}$$

In Equation (14), $c_i$ is the flow size of the flow in the transmission interval from $p'_i$ to $q'_i$. In Equation (15), $u$ is the sending node of the flow, $v$ is the receiving node of the flow, $\Gamma(u)$ is the set of neighbor nodes of node u, $f_i^{uv}$ is the flow sent by node $u$, and $f_i^{vu}$ is the flow received by node $v$.

### 3. Energy-Saving Routing Architecture under SDN

SDN is a new type of network architecture, it decouples control and forwarding from each other and supports centralized and programmable network control. Based on the advantages of SDN's fast and flexible network control capability, customizable network infrastructure, and low network control and management costs, our method introduces an AI plane based on the data plane and control plane of SDN architecture for efficient network policy generation and globalized, real-time and customized network control management, which realizes real-time network traffic monitoring and identification elephant flows/mice flows under energy efficient traffic scheduling. The architecture of energy-saving routing traffic scheduling under SDN is shown in Figure 1.

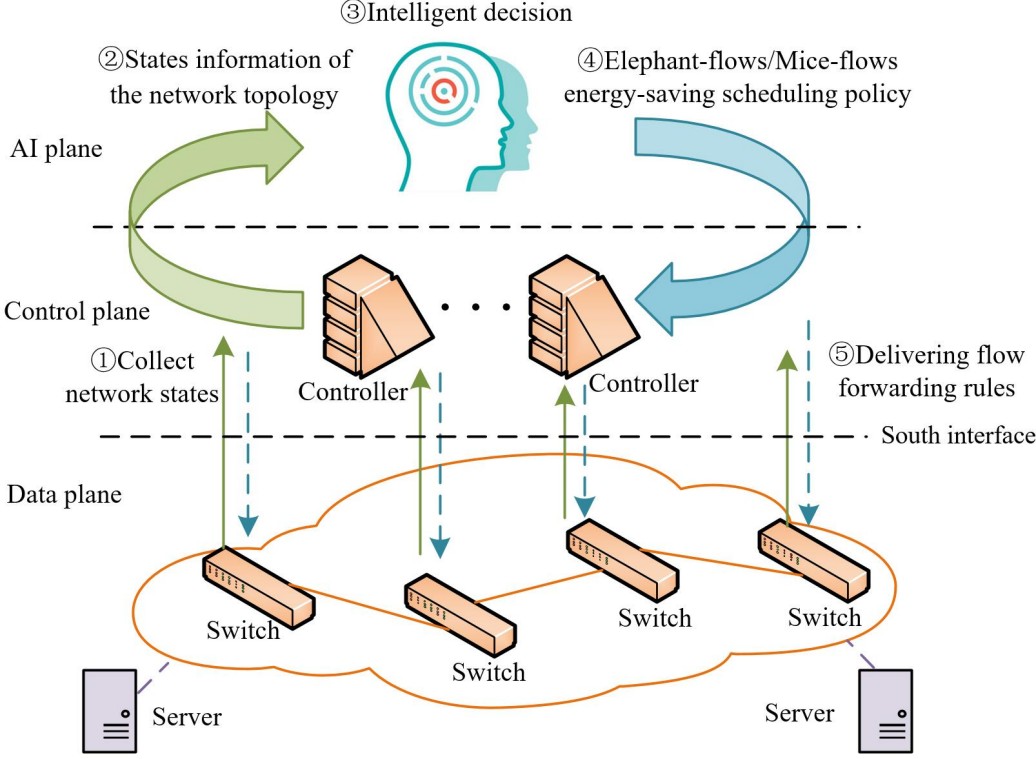

**Figure 1.** Energy-saving routing traffic scheduling architecture under SDN.

The main functions of the data plane, control plane, and AI plane in the figure are as follows: (1) The data plane is composed of hardware devices such as switches and servers, and is mainly responsible for data forwarding between network devices. (2) The control plane provides several centralized logic controllers, which are mainly responsible for functions such as dynamic programming and control of forwarding network resources, effectively reducing the burden of distributed network control and management. Among them, the network detection module located in the control plane can periodically collect the link bandwidth, delay, throughput, and network traffic information data in the network topology through the south interface (openflow protocol) on a regular basis, and effectively monitor the characteristics of network flows (elephant flows/mice flows). If the current

traffic demand bandwidth exceeds 10% of the link bandwidth, the flow is determined to be an elephant flow, otherwise, it is a mice flow. (3) The AI plane is located above the control plane as a decision-making module, which can combine with the real-time network status and traffic characteristics. After neural network training, energy-saving paths can be adapted for elephant flows and mice flows, and an intelligent network policy can be generated so that the controller can generate the flow table according to the path policy.

## 4. Energy-Saving Routing Scheme Based on DDPG

### 4.1. Ee-Routing Algorithm Framework

Our method realizes energy-saving traffic scheduling based on the environment perception and deep learning decision-making capabilities of deep reinforcement learning. The Ee-Routing algorithm framework uses DDPG from a novel deep reinforcement learning method, which constructs an actor-critic framework by combining the DQN method with the DPG method, using neural networks instead of policy functions and Q functions to form efficient and stable discrete action control models [23]. In this paper, the DDPG algorithm in deep reinforcement learning is introduced into the energy-saving traffic scheduling process, and the advantages of DDPG's online network and target network, as well as the application of the soft update algorithm, are used to promote a more stable learning process and ensure model convergence; DDPG needs fewer samples and there is no need to integrate the action space, which effectively reduces the complexity of the algorithm. This method replaces the traditional neural network in DDPG with CNN, integrates the CNN update process with the online network and target network in DDPG, and utilizes the advantages of CNN in high-dimensional data processing, which can effectively speed up the algorithm convergence efficiency. The energy-saving routing framework based on DDPG is shown in Figure 2.

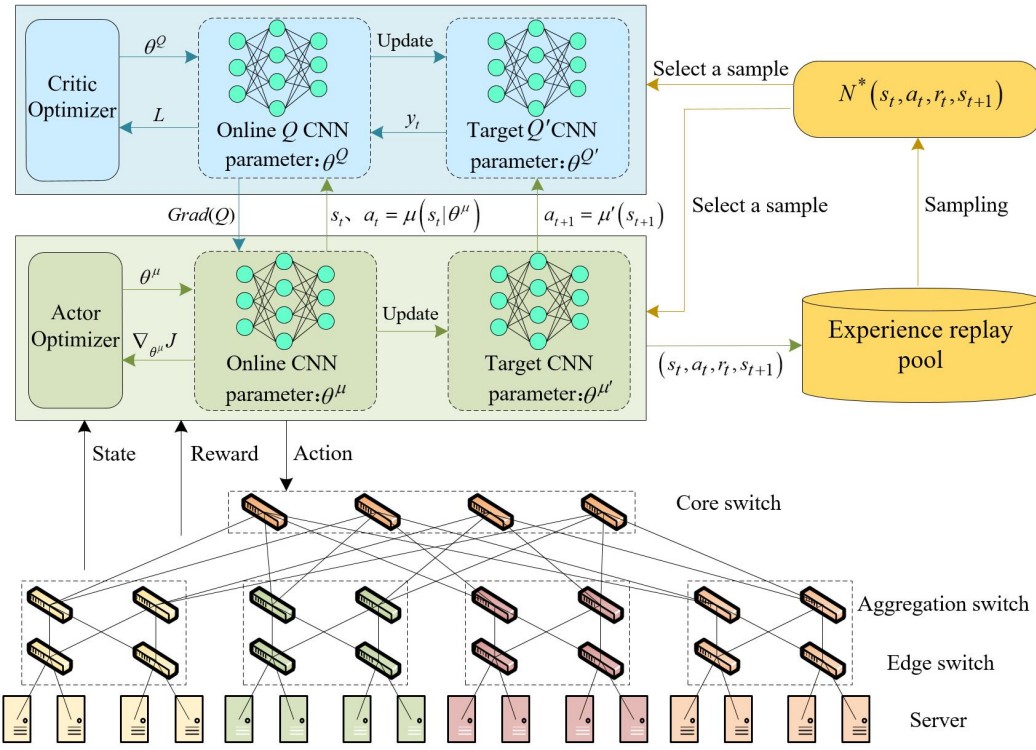

**Figure 2.** Energy-saving routing framework based on DDPG.

Figure 2 describes the update process of the actor-critic online network and target network, as well as the interaction between the actor-critic and the environment. The specific actor-critic update process is as follows:

(1) Update the online network: The online network consists of the actor online network and the critic online network, in which the actor online network can generate the current action $a_t = \mu(s_t|\theta^\mu)$ according to the current state $s_t$ and random initialization parameter $\theta^\mu$, and interact with the environment to obtain the reward $r_t$ and the next state $s_{t+1}$. The combination of $s_t$ and $a_t$ is input to the online critic network, the current action value function $Q(s_t, a_t|\theta^Q)$ is generated by the online critic iteration, and $\theta^Q$ is a random initialization parameter. The critic online network provides gradient information $grad[Q]$ for the actor online network to help the actor online network update the network, and the calculation process of the actor online network gradient is shown in Equation (16).

$$\nabla_{\theta^\mu} J = grad[Q] * grad[\mu] \approx \frac{1}{N} \sum_t \nabla_a Q(s, a|\theta^Q)|_{s=s_t, a=\mu(s_t)} \nabla_{\theta^\mu} \mu(s|\theta^\mu)|s_t \qquad (16)$$

Among them, $grad[Q]$ is provided by the critic online network to ensure that the actor online network action direction obtains a higher reward, and $grad[\mu]$ is provided by the actor online network to ensure that the action of actor online network obtains a higher reward. In addition, the critic online network can update the network parameters through the error equation to minimize the calculation error, and the calculation process is shown in Equation (17).

$$L = \frac{1}{N} \sum_t \left( y_t - Q(s_t, a_t|\theta^Q) \right)^2 \qquad (17)$$

where $y_t$ is the target reward sought by the critic target network.

(2) Update the target network: In order to ensure the effectiveness and convergence of network training, the DDPG framework provides the actor target network and the critic target network with the same structure as the online network. The actor target network selects the next state $s_{t+1}$ from the experience replay pool, and obtains the next optimal action $a_{t+1} = \mu'(s_{t+1})$ after iterative training. The network parameter $\theta^{\mu'}$ is obtained by periodically copying the actor online network parameter $\theta^\mu$, the action $a_{t+1}$ and the state $s_{t+1}$ are combined input to the critic target network, the the critic target network is iteratively trained to obtain the target value function $Q'\left(s_{t+1}, \mu'\left(s_{t+1}|\theta^{\mu'}\right)|\theta^{Q'}\right)$, and the parameter $\theta^{Q'}$ is obtained by periodically copying the actor online network parameter $\theta^Q$. The calculation process of the critic target network providing the target return value $y_t$ for the critic online network is shown in Equation (18).

$$y_t = r_t + \gamma Q'(s_{t+1}, \mu'(s_{t+1}|\theta^{\mu'})|\theta^{Q'}) \qquad (18)$$

(3) Experience replay pool $D$: The idea of experience replay is used to store information such as state, action, and reward during the interaction between the agent and the environment, and the valid information is transferred and stored in the tuple $(s_t, a_t, r_t, s_{t+1})$ as a sample, and priority is assigned to each sample. In order to avoid correlation between the training data, the sample data can be sampled for error value calculation based on the priority when updating the policy.

### 4.2. Ee-Routing Neural Network Interaction with Environment

In the Ee-Routing algorithm framework, a CNN is used for the neural network training process of DDPG. A CNN is a deep network architecture with strong discriminative ability, its structure mainly includes multi-layer convolution and nonlinear activation functions, and widely used in the field of image recognition, and has a good training effect on the input of high-dimensional complex data. According to the local receptive field, shared weight, and pooling ability in the CNN training process, Ee-Routing inputs the network state features per unit time into the CNN convolution kernel and performs the convolution operation on the input network state to obtain the features map. The features map of the previous layer is compressed to achieve dimensionality reduction processing, which helps the training process to converge faster. After multi-layer convolution and pooling

operations, the final full connection layer outputs a set of path weights. The agent based on the CNN network architecture interacts with the network environment repeatedly, continuously uses the network state and accumulated rewards to update the action, and finally achieves energy-saving traffic scheduling. The SAR settings of the interaction process between the agent and the environment are as follows.

(1)    State space

This method takes data plane energy-saving and network performance as the joint optimization goal, which is mainly related to the state's information on link transmission rate, link utilization rate, and link energy consumption at the current moment and historical moment, assuming that there are $m$ links. In this method, the three state features are jointly used as the state set $state_t = \{s_{LTR_t}, s_{LUR_t}, s_{LEC_t}\}$ to input the neural network for training, and the state elements in the state set are respectively mapped to a state feature of the CNN. The state feature map is shown in Figure 3. Among them, the transmission rate of these links in the network is recorded as $s_{LTR_t} = \{ltr_1(t), ltr_2(t), \cdots ltr_m(t)\}$, and input channel1 as the state features. The utilization rate of these links is recorded as $s_{LUR_t} = \{lur_1(t), lur_2(t), \cdots lur_m(t)\}$, and input channel2 as the state features. The energy consumption of these links is recorded as $s_{LEC_t} = \{lec_1(t), lec_2(t), \cdots lec_m(t)\}$, and input channel 3 as the state features.

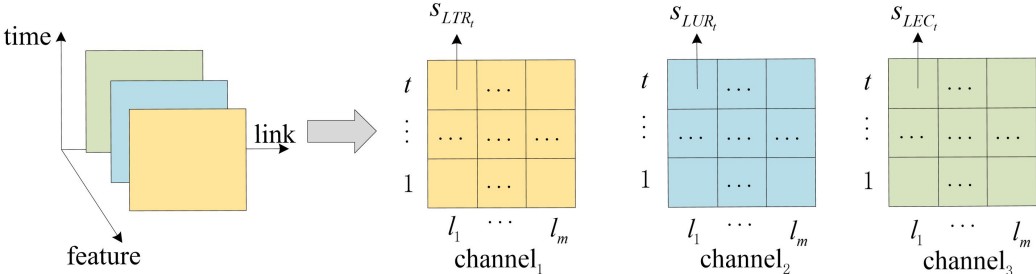

**Figure 3.** State feature mapping.

(2)    Action space

According to the network state and reward feedback information, our method set the action as the comprehensive weight of the performance and energy consumption of each path, which makes the flows transmit uniformly in time and space. The specific action set is shown in Equation (19).

$$action = \{a_{w1}, a_{w2}, \cdots a_{wi}, \cdots, a_{wn}\} \quad wi \in W \tag{19}$$

Among them, $W$ is the set of paths that network traffic can transmit, $wi$ represents the $wi - th$ path in the optional transmission path set, and $a_{wi}$ represents an action value in the action set, which refers to the path weight value of the $wi - th$ path. Since this method divides the flows into elephant flows and mice flows for traffic scheduling, if the controller detects that the network flow is an elephant flow, it adopts a multi-path mode for traffic transmission and allocates traffic according to the proportion of different path weights to the total path weights. For example, if there are $n$ optional paths between a source node $s_i$ and a target node $d_i$, the traffic distribution proportion of each path sent from source node $s_i$ to target node $d_i$ can be calculated by Equation $Proportion_i = \frac{a_{wi}}{\sum\limits_{i=1}^{n} a_{wi}}$. If the controller detects that the network flow is a mice flow, then a single-path mode is adopted for traffic transmission, and the path with the larger path weight is selected as the flow transmission path, and the maximum path weight can be selected as the mice-flow transmission path through the set $\{a_{w1}, a_{w2}, \cdots a_{wi}, \cdots, a_{wn}\}$.

(3)  Reward function

Considering the characteristics of different flows, this method sets the reward functions of the elephant flows and the mice flows. The main optimization goals of the elephant flows are low energy consumption, low packet loss rate, and high throughput, therefore, the normalized values of energy consumption, packet loss rate, and throughput are used as reward factors. In order to intuitively experience the cumulative rewards, the inverse of the energy consumption and the inverse of the packet loss rate are selected as the reward value factors when the reward function is set so that the optimization goal performance and the rewards increase in direct proportion. The specific calculation process is shown in Equation (20).

$$reward_{elephent} = \eta \frac{1}{Power_{total}'} + \tau \frac{1}{Loss_{elephent}'} + \rho Throught_{elephent}' \tag{20}$$

In Equation (20), $\eta$, $\tau$, and $\rho$ are all between 0 and 1, which can be based on the proportion of the importance of energy consumption, packet loss rate and throughput in the elephant flows. In the same way, the mice flows take low energy consumption, low packet loss rate, and low delay as the optimization goals, and take the normalized inverse of the three as the reward factors. The specific calculation process is shown in Equation (21).

$$reward_{mice} = \eta \frac{1}{Power_{total}'} + \tau \frac{1}{Loss_{mice}'} + \rho \frac{1}{Delay_{mice}'} \tag{21}$$

*4.3. Implementation of Ee-Routing Algorithm*

In order to ensure the optimization effect of energy-saving traffic scheduling, the Ee-Routing algorithm uses different neural networks for training on elephant flows and mice flows. The overall algorithm process of Ee-Routing is shown in Algorithm 1. Lines 1–10 of the algorithm add the judgment of the elephant flow and the mice flow to the new flow of the network and input the elephant flow state and the mice flow state into the $CNN_1$ and $CNN_2$, respectively, based on the different cumulative rewards of the elephant flow and mice flow in Section 4.2, the path weights of elephant flow and mice flow are output, and the SDN controller deploys the optimized paths of elephant flow and mice flow according to the weight information. Lines 11–23 of the algorithm design the parameter update process of the network flows during the training process. First, the agent in line 12 obtains the initialization state $s_1$ from the environment, and then lines 14–17 perform actions according to the current network state to obtain the reward and the action at the next moment and save $(s_t, a_t, r_t, s_{t+1})$ into the experience replay pool $D$, and finally randomly select mini-batch training samples from $D$ through lines 18–23 to complete the actor-critic network training. After several rounds of iterations, the Ee-Routing algorithm is completed, and the path weights of the elephant flow and mice flow are output, which realizes the optimization of energy-saving routing traffic scheduling.

---

**Algorithm 1** Ee-Routing Algorithm Process

---

1: Random initialize Parameter $\theta^\mu$, $\theta^Q$, $\theta^{\mu'}$, $\theta^{Q'}$ and $D$
2: Input: Network topology state $state_t = \{s_{LTR_t}, s_{LUR_t}, s_{LEC_t}\}$
3: output: Path weights of $w_{elephent}$ and $w_{mice}$
4:    While new flow do
5:        Determine the type of flow
6:        If $Flow_{elephent}$ then
7:            Input $s_t$ into $CNN_1$ and output $w_{elephent}$
8:            Install the flow rules according elephant-flow scheduling
9:        Else
10:            Input $s_t$ into $CNN_2$ and output $w_{mice}$
11:            Install the flow rules according mice-flow scheduling
12:        End if
13:    End while
14:    For episode = 1, $M$ do
15:        Initalize state $s_1$
16:        For $t = 1$, $T$ do
17:            Select action $a_t = \mu(s_t|\theta^\mu)$ according to the current policy
18:            Execute $w_t \leftarrow a_t = \mu(s_t|\theta^\mu)$ made the SDN controller builds a route path
19:            Obtain $r_t$ and $s_{t+1}$
20:            Store transition $(s_t, a_t, r_t, s_{t+1})$ in $D$
21:            Sample a random mini batch of $N^*(s_t, a_t, r_t, s_{t+1})$ from $D$
22:            $y_t = r_t + \gamma Q'(s_{t+1}, \mu'(s_{t+1}|\theta^{\mu'})|\theta^{Q'})$
23:            $L = \frac{1}{N} \sum_t (y_t - Q(s_t, a_t|\theta^Q))^2$
24:            $\nabla_\theta \mu J \approx \frac{1}{N} \sum_t \nabla_a Q(s, a|\theta^Q)|_{s=s_t, a=\mu(s_t)} \nabla_{\theta^\mu} \mu(s|\theta^\mu)|_{s_t}$
25:            $\theta^{Q'} \leftarrow \tau\theta^Q + (1-\tau)\theta^{Q'}$
26:            $\theta^{\mu'} \leftarrow \tau\theta^\mu + (1-\tau)\theta^{\mu'}$
27:        End for
28:    End for

---

## 5. Experimental Evaluation

### 5.1. Experimental Environment and Parameter Configuration

In the experimental process, the network simulation software Mininet [24] is used to complete the Ee-Routing intelligent routing algorithm for performance testing. The experiment uses the Fat Tree [25] data center network topology, including 20 switches with 4 ports, 16 servers, and 48 links. The link bandwidth is set to 100 Mbps by default. To simulate the network traffic of the data center, 80% of the flows are set to mice flows, and 20% of the flows are set to elephant flows. Ee-Routing is based on the SDN network architecture and uses the DDPG algorithm framework to implement routing updates. The control plane uses the RYU controller to implement centralized network management, and the data plane uses the Open vSwith virtual switch to implement the networking of the data plane. The experimental software environment is Linux operating system Ubuntu18.04, Tensorflow1.8.0, and Python3.5.0, the experimental hardware implementation platform is i5-10600KF-CPU, 16GB-DDR4 memory, and two GTX-1080 8 G graphics cards.

During the Ee-Routing training process, the neural network uses the Adam optimizer and the Relu activation function. Its parameters involve the number of algorithm training steps, the learning rate, the target network parameter update rate, and the size of the experience replay pool during the DRL training process. The specific configuration is shown in Table 1.

**Table 1.** Simulation experiment parameters configuration.

| Experimental Parameters | Parameter Value |
|---|---|
| training steps $T$ | 80.000 |
| learning rate of actor/critic $lr$ | 0.002/0.001 |
| target network parameter update rate $tau$ | 0.001 |
| size of the experience replay pool $D$ | 4500 |
| training steps of the experience replay pool $M$ | 150 |
| discount factor $\lambda$ | 0.7 |
| greedy $\varepsilon$ | 0.01 |
| momentum $m$ | 0 |
| raining batch size $bath_size$ | 128 |
| reward weight parameter $\alpha$ $\beta$ $\gamma$ | 0–1 |

*5.2. Experimental Comparison*

In order to evaluate the energy-saving and network performance advantages of the proposed routing algorithm Ee-Routing, the experiment compares Ee-Routing with the current optimal energy-saving routing algorithm, high-performance intelligent routing algorithm, and heuristic energy-saving routing algorithm. The comparison algorithms include: (1) Time Efficient Energy Aware Routing (DQN-EER) [15]; (2) Deep Q-Network-based Energy-Efficient Routing (EARS) [12]; (3) intelligence-driven experiential network architecture for automatic routing in software-defined networking (TEAR) [26]. The main comparison contents include algorithm convergence speed and energy-saving effect, as well as performance indicators such as network average end-to-end delay, throughput, and packet loss rate. Among them, the energy-saving effect evaluation index is shown in Equation (22).

$$Power_{save} = 1\text{-} \frac{lec_i}{lec_{full}} \times 100 \tag{22}$$

In Equation (22), $lec_i$ represents the energy consumption of these links consumed by the current routing algorithm, and $lec_{full}$ is the total energy consumption of these links when the link is fully loaded.

5.2.1. Convergence of the Algorithm

In order to verify the convergence of Ee-Routing, the experiment takes network energy consumption, delay, throughput, and packet loss rate as the optimization goal, and maximizes the cumulative rewards as the convergence evaluation standard. The reward functions parameter weights $\eta$, $\tau$, and $\rho$ are all set to 1, $\alpha$ is set to 2 and $\mu$ is set to 1 in the energy consumption function. Since TEAR is a traditional routing algorithm without convergence, this experiment only compares the convergence of the intelligent routing algorithm Ee-Routing with EARS and DQN-EER. The experimental results are shown in Figure 4. It can be seen from the figure that as the number of training steps increases, the cumulative rewards of Ee-Routing gradually stabilize around $30 \times 10^3$ steps, while both EARS and DQN-EER tend to stabilize around $50 \times 10^3$ steps. Therefore, the upward trend of Ee-Routing is the most obvious. The DQN-EER training process uses DQN as the algorithm framework, it adopts a random policy method. There is a lot of variability in the output action so the parameter update direction is not necessarily the optimal direction of the policy gradient. Therefore, the DQN-EER algorithm has poor convergence. The EARS training process uses DDPG as the algorithm framework. On the basis of DQN, the online network and target network are added to learn deterministic behavior policies. The process of integrating actions is reduced, and the derivative of the reward function to actions is increased. The convergence efficiency of the EARS algorithm is accelerated, but the EARS training process only uses traditional neural networks, resulting in convergence efficiency still needing to be improved. Ee-Routing improves the online network and target network in DDPG based on CNN, takes advantage of the CNN's local perception and parameter

sharing, has the advantage of processing high-dimensional data, and can further accelerate the algorithm convergence efficiency. Therefore, Ee-Routing is guaranteed to have good convergence.

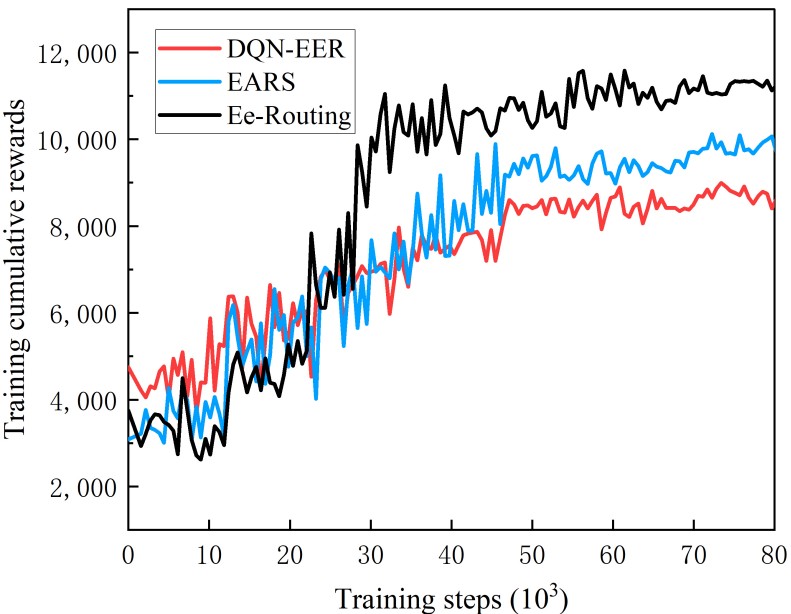

**Figure 4.** Changes in cumulative rewards.

### 5.2.2. Energy-Saving Comparison

In order to verify the energy-saving effect of Ee-Routing in real network scenarios, the experiment sets up network load environments with different traffic intensities. The experiment takes network energy consumption, delay, throughput, and packet loss rate as optimization goals, and in order to highlight the importance of energy-saving, the reward function parameter weight $\eta$ is set to 1, $\tau$ and $\rho$ are both set to 0.5, and $\alpha$ is set to 2 and $\mu$ is set to 1 in the energy consumption function, and the traffic intensity is set to 20%, 40%, 60%, and 80%. Ee-Routing is compared with TEAR, EARS, and DQN-EER. The experimental results are shown in Figures 5–8. As can be seen from the figure, with the increase in traffic intensity, Ee-Routing's energy-saving effect is better than other routing algorithms. Among them, TEAR is a traditional energy-saving routing algorithm, and the energy-saving effect does not change with the increase of training steps. Therefore, the energy-saving effect is obviously weakened under the condition of increasing traffic intensity. DQN-EER as an intelligent routing algorithm is more suitable for complex network environments than TEAR, therefore, with the increase in the number of training steps, the energy-saving effect is more obvious when the traffic intensity is large, but the DQN-EER training process adopts the DQN framework and traditional neural network, and the optimization policy direction is random, resulting in the energy-saving convergence efficiency and energy-saving effect still needing to be improved. Due to the lack of consideration of energy-saving indicators in the EARS intelligent routing algorithm, with the increase in the number of training steps, the improvement of energy saving effect is small, especially when the flow intensity is large, and the energy saving trend is relatively gentle. On the basis of DQN-EER and EARS, Ee-Routing considers energy saving and network performance at the same time, and based on the improved DDPG of GNN for training and updating parameters, using the deterministic policy of DDPG, and the advantages of CNN local perception and parameter sharing, Ee-Routing has the most obvious energy-saving effect compared to other routing algorithms. After the algorithm model training tends to be stable, compared with the intelligent routing algorithm DQN-EER with better energy-saving, the energy-saving percentage is increased by 13.93%.

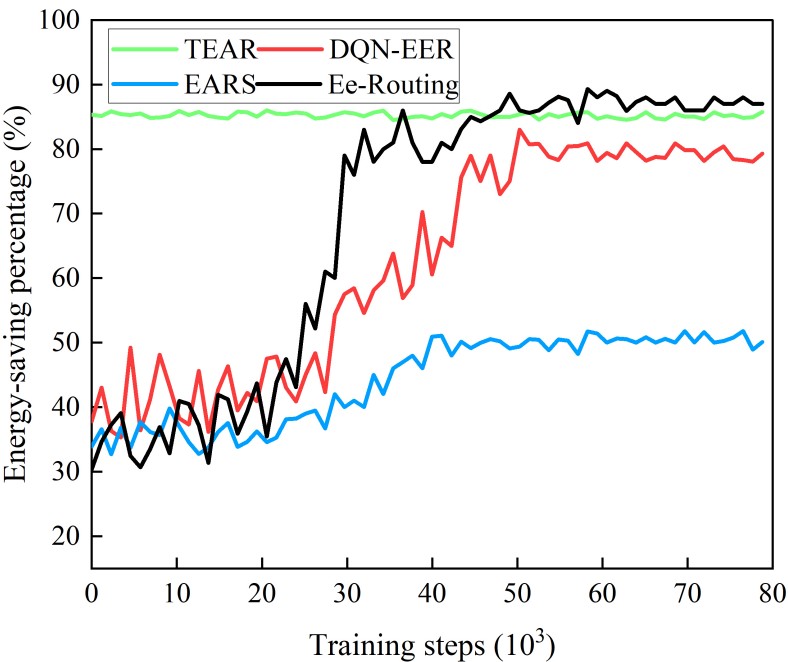

**Figure 5.** Energy-saving percentage under 20% traffic intensity.

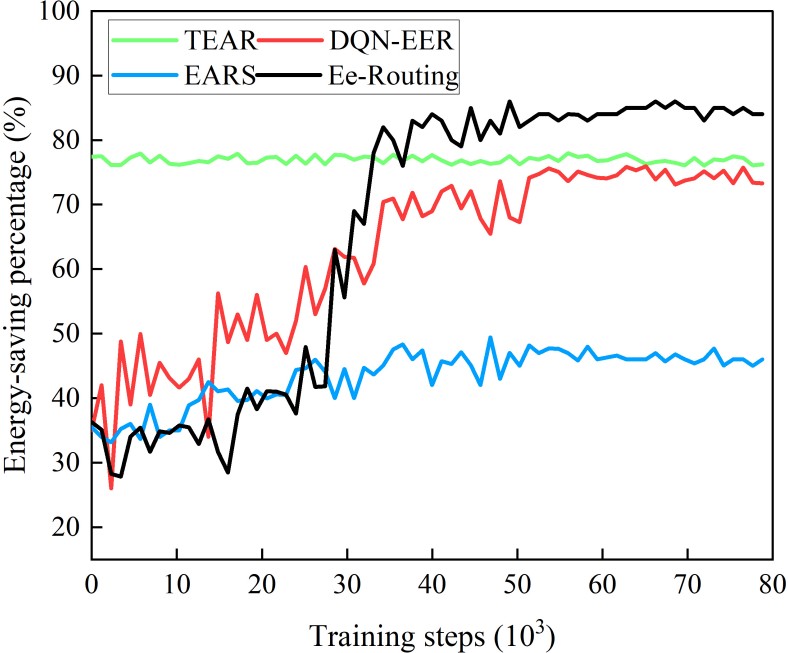

**Figure 6.** Energy-saving percentage under 40% traffic intensity.

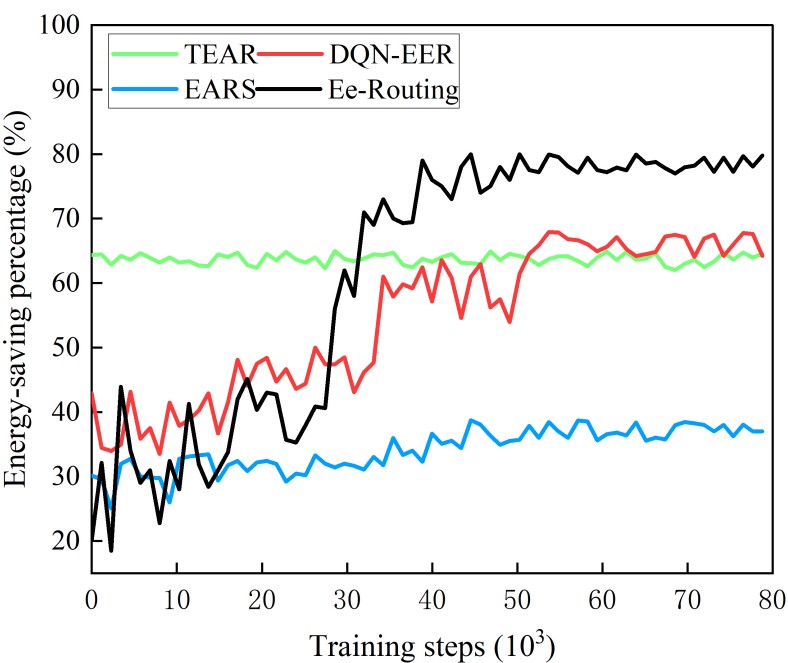

**Figure 7.** Energy-saving percentage under 60% traffic intensity.

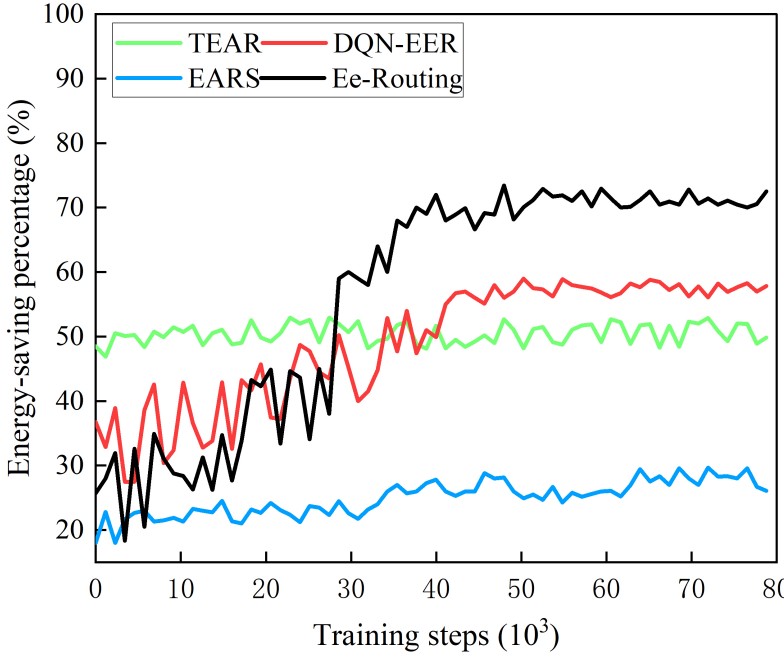

**Figure 8.** Energy-saving percentage under 80% traffic intensity.

5.2.3. Performance Comparison

In order to verify the network performance indicators of Ee-Routing in real network scenarios, the experiment sets up network load environments with different traffic intensities. The experiment takes network energy consumption, delay, throughput, and packet loss rate as optimization goals, the reward function parameter weight $\eta$ is set to 0.5, $\tau$ and $\rho$ are both set to 1, $\alpha$ is set to 2, and $\mu$ is set to 1 in the energy consumption function, and the traffic intensity is set to 20%, 40%, 60%, and 80%. Ee-Routing is compared with TEAR, EARS, and DQN-EER. The experimental results are shown in Figures 9–11. As can be seen from the figures, with the increase in traffic intensity, Ee-Routing is better than other routing algorithms in delay, throughput, and packet loss rate. Among them, the traditional routing algorithm TEAR is often difficult to adapt to the optimal path according to the traffic load

with the increase of traffic intensity, which easily causes link congestion. The DQN-EER intelligent routing algorithm takes the energy-saving of the data plane and the delay of the control plane as evaluation indicators, compared with TEAR, which improves the load balancing ability. EARS, as a high-performance intelligent routing algorithm, considers the delay, throughput, and load balancing rate under different traffic intensities, and adopts the DDPG algorithm framework, so it has better load balancing ability than DQN-EER. Ee-Routing uses delay, throughput, and packet loss rate as network performance evaluation indicators, adopts the method of different scheduling between elephant flows and mice flows, and improves the DDPG algorithm framework based on CNN to ensure Ee-Routing has better energy-saving traffic scheduling advantages in complex network environments, which makes Ee-Routing have the best load balancing ability compared with the above three routing algorithms, and compared with the intelligent routing algorithm EARS with better performance, the delay is reduced by 13.73%, the throughput is increased by 10.91%, and the packet loss rate is reduced by 13.51%.

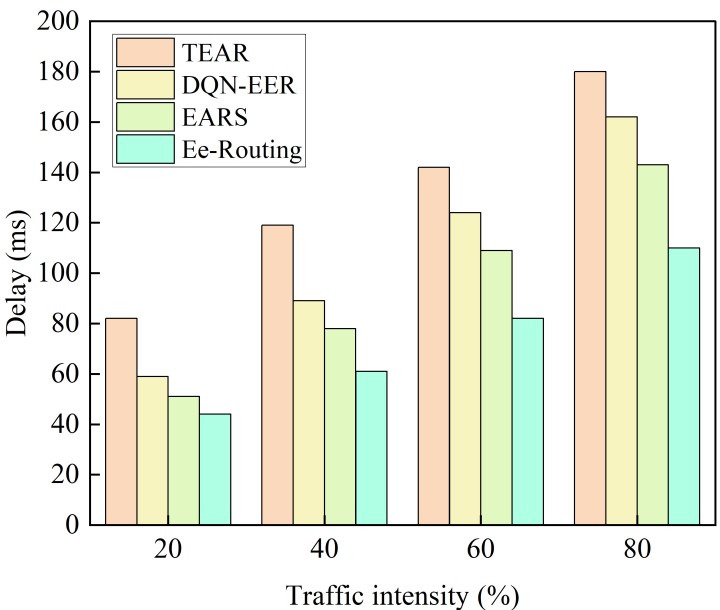

**Figure 9.** Delay comparison under different traffic intensities.

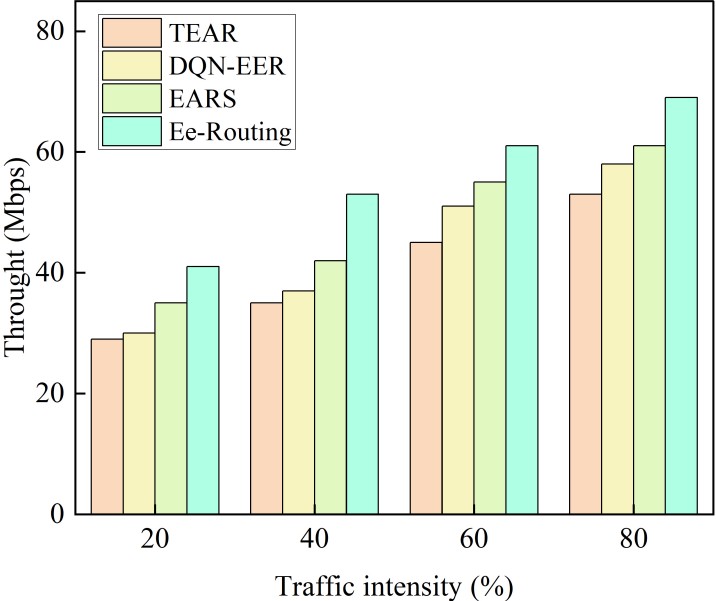

**Figure 10.** Throughput comparison under different traffic intensities.

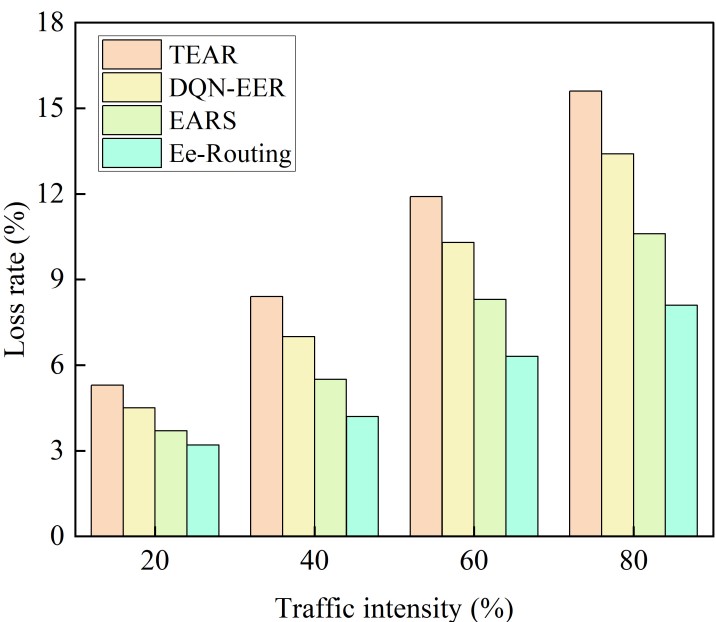

**Figure 11.** Packet loss rates comparison under different traffic intensities.

## 6. Conclusions

In this paper, we propose an energy-saving routing algorithm, Ee-Routing, based on deep reinforcement learning, which uses DDPG and CNN to dynamically perceive complex and changeable network environments, it achieves two goals, one is the convergence and stability of Ee-Routing, and the other is that Ee-Routing has better energy saving and network performance advantages under different traffic intensities. In this paper, the Ee-Routing routing algorithm is compared with the TEAR, EARS, and DQN-EER routing algorithms. The results show that Ee-Routing has good convergence and stability. Compared with the DQN-EER routing algorithm, the energy saving percentage of Ee-Routing is increased by 13.93%, and compared with the EARS routing algorithm, Ee-Routing reduces the delay by 13.73%, increases the throughput by 10.91%, and reduces the packet loss rate by 13.51%.

Overall, the conclusion of this paper shows that the Ee-Routing routing algorithm has good energy saving and network performance advantages, and can be applied to the network environment optimization of various data centers. Whether it is elephant flows or mice flows, the Ee-Routing routing algorithm can effectively reduce network energy consumption and improve network performance, it lays the foundation for further network energy saving and performance optimization. Nevertheless, this paper mainly considers the energy-saving and network performance of the data plane in the SDN architecture and does not consider the network energy consumption and network performance of the control plane. SDN is a new network architecture with decoupling of data forwarding and logical control, with the sharp increase of network users and network traffic, the network energy consumption and network performance of the control plane will also be affected. Therefore, the energy-saving and network performance of the SDN control plane also have great research significance. In the following research work, we will further study the network energy consumption and network performance of the control plane in SDN.

**Author Contributions:** Conceptualization, X.Z. and W.H.; methodology, X.Z. and S.W. and J.Z.; software, X.Z. and H.Z.; validation, X.Z., W.H., S.W., J.Z. and H.Z.; formal analysis, X.Z. and W.H.; investigation, X.Z., J.Z. and H.Z.; data analysis, X.Z., S.W. and H.Z.; writing—original draft preparation, X.Z. and W.H.; writing—review and editing, X.Z., W.H., S.W., J.Z. and H.Z. All authors have read and agreed to the published version of the manuscript.

**Funding:** This research was funded in part by the National Natural Science Foundation of China (62002382, 62072416), in part by the Project of Science and Technology in Henan Province (222102210175, 222102210111), and in part by the Postgraduate Education Reform and Quality Improvement Project of Henan Province (YJS2022AL035).

**Conflicts of Interest:** The authors declare no conflict of interest.

## Abbreviations

The following abbreviations are used in this manuscript:

QoS   Quality of Service
QoE   Quality of Experience
SDN   Software-Defined Network
DDPG  Deep Deterministic Policy Gradient
CNN   Convolutional Neural Network
DQN   Deep Q Network
DPG   Deterministic Policy Gradient
LTR   Link Transmission Rate
LUR   Link Utilization Rate
LEC   Link Energy Consumption

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
