# Peer review of "Research on Energy-Saving Routing Technology Based on Deep Reinforcement Learning"

_electronics, doi:10.3390/electronics11132035_

Round 1

Reviewer 1 Report

The paper deals with energy saving which is an important and up-to-date subject.  The authors proposed a deep learning algorithm that allows for dynamic changes in network environments.

The paper is well structured and the results support the conclusions. The very useful Figures 9, 10 and 11 show the quality of the algorithms.

In my opinion, the paper can be published in the Journal. However, I would have one remark. It is worthy to add a table of the abbreviations used. It would improve the readability of the paper.

Reviewer 2 Report

The paper has a good potential for being appreciated and cited, but it requires some improvements and also extensions.

Highlight the research gap in existing research and in literature. The importance of the proposed integrated approach with respect to the problem statement should have been in focus.

About the related works, further papers should be added to the literature review. Each paper should clearly specify what is the proposed methodology, novelty, and results with experimentation. At the end of related works, highlight in some lines what overall technical gaps are observed in existing works, that led to the design of the proposed approach. To better delineate the context and the different possible solutions, you can consider the following papers as references: https://ieeexplore.ieee.org/abstract/document/9409962 and https://www.mdpi.com/2079-9292/10/18/2250.

The future scope of the methodology should be extended/highlighted. Improve the conclusion, and clarify the conclusion of this article with its significance for follow-up research.
